# KALEIDO: OPEN-SOURCED MULTI-SUBJECT REFERENCE VIDEO GENERATION MODEL

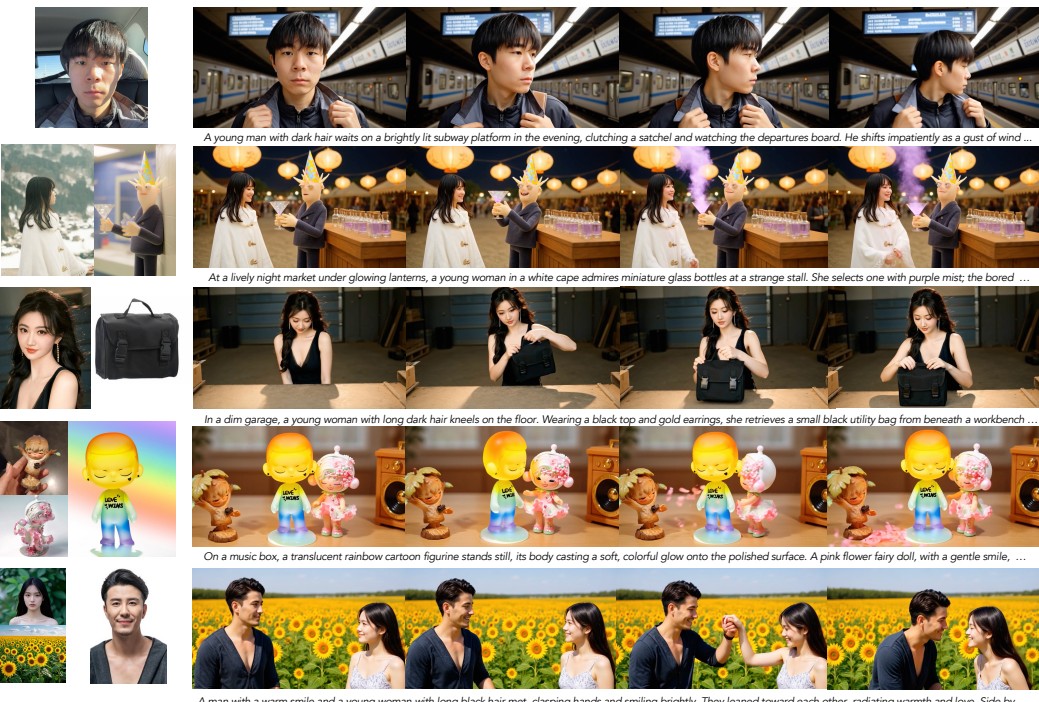

Figure 1: Subject-to-video generation by *Kaleido* covering humans, objects, and controlled backgrounds in both single and multi-subject cases.

## ABSTRACT

We present *Kaleido*, a subject-to-video (S2V) generation framework, which aims to synthesize subject-consistent videos conditioned on multiple reference images of target subjects. Despite recent progress in S2V generation models, existing approaches remain inadequate at maintaining multi-subject consistency and at handling background disentanglement, often resulting in lower reference fidelity and semantic drift under multi-image conditioning. These shortcomings can be attributed to several factors. Primarily, the training dataset suffers from a lack of diversity and high-quality samples, as well as cross-paired data, i.e., paired samples whose components originate from different instances. In addition, the current mechanism for integrating multiple reference images is suboptimal, potentially resulting in the confusion of multiple subjects. To overcome these limitations, we propose a dedicated data construction pipeline, incorporating low-quality sample filtering and diverse data synthesis, to produce consistency-preserving training data. Moreover, we introduce Reference Rotary Positional Encoding (R-RoPE) to process reference images, enabling stable and precise multi-image integration. Extensive experiments across numerous benchmarks demonstrate that Kaleido significantly outperforms previous methods in consistency, fidelity, and generalization, marking an advance in S2V generation. The source code and trained model checkpoints for this study are available at *here*.

# 1 INTRODUCTION

Recent years have witnessed rapid and highly promising advances in video generation. Inspired in part by the success of Sora, diffusion models integrated with Diffusion Transformers (DiT) (Peebles & Xie, 2023; Esser et al., 2024) have emerged as a prevailing paradigm and get further developed. Commercial models like Veo3 (DeepMind) and Kling (Kuaishou) have already achieved video quality on par with professional production standards, introducing a new workflow paradigm for video content creation that greatly improves efficiency while reducing production costs. In the open-source domain, models such as Wan (Wang et al., 2025) and CogVideoX (Yang et al., 2025) not only share these advantages, but also facilitate customized fine-tuning for specific applications.

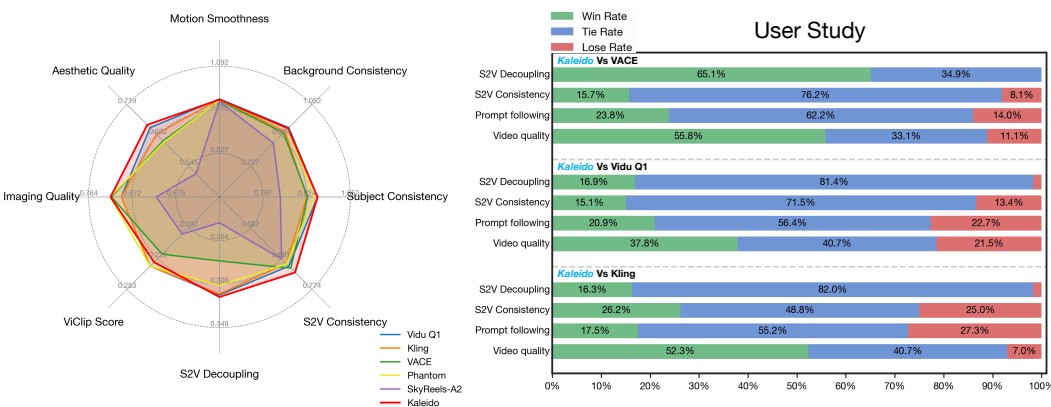

Figure 2: Subject-to-video evaluation (left) and user study results comparing *Kaleido* with VACE, Kling, and Vidu-Q1 (right).

Current pretraining video generation research primarily focuses on two major tasks: text-to-video (T2V) and image-to-video (I2V) generation. The former synthesizes videos directly from textual descriptions, often yielding content with high creative diversity but also considerable randomness. The latter transforms a static image into a dynamic video, imposing a strict constraint of identical first frame, which tends to limit creative flexibility. Consequently, the need for more flexible control over video generation has grown increasingly strong. **Subject-to-video (S2V) generation**, which aims to synthesize subject-consistent videos conditioned on multiple reference images of target subjects, has attracted rising attention. Commercial systems such as Vidu (Bao et al., 2024) and Kling (Kuaishou) exemplify this trend, demonstrating substantial potential in industries such as e-commerce and advertising.

The S2V task requires decoupling target subjects from given reference images and generating videos according to textual prompts, while maintaining the appearance of the subjects consistently. The subjects encompass a wide range of visual entities, including humans, objects, and backgrounds. It unifies the creativity of T2V generation and the controllability of I2V generation, enabling more flexible control in video generation.

However, existing open-source S2V models remain inferior to their closed-source counterparts. This gap is primarily reflected in the difficulty of maintaining consistent visual appearances across diverse subject compositions, as well as in producing high-quality videos. This performance gap can be primarily attributed to two fundamental challenges:

- **Lack of effective training data.** In most recent data construction pipelines, reference images are naively selected from video frames. Models trained on such data often tend to completely replicate the subjects in reference images (without altering their viewpoints, poses, or other dynamic attributes), rather than decoupling the subjects and focusing on their intrinsic characteristics. Consequently, generated videos may inherit extraneous elements, such as superfluous background details or irrelevant objects present in the reference images, which are usually undesirable in videos. Furthermore, existing models often struggle to preserve satisfactory consistency in various scenarios, including multi-subject

compositions or scenes involving animated characters. This limitation is largely due to the insufficient coverage and quality of training data.

- **Inadequate conditioning strategies.** Current strategies for incorporating reference image information into the generation pipeline are generally suboptimal, hindering the model's ability to efficiently capture and represent the subject's characteristics. For example, Phantom (Liu et al., 2025) adopts latent feature concatenation along the sequence dimension, while this method may cause different reference objects to overlap spatially, leading to undesirable compositional artifacts. VACE (Jiang et al., 2025) employs an adapter-based architecture, but it needs a non-negligibly additional inference cost.

We introduce a set of methods that allow open-source models to attain performance on par with closed-source counterparts. Our contributions can be summarized as follows:

- **A comprehensive data construction pipeline.** Our data pipeline employs multi-class sampling, stringent filters, and cross-paired data construction to enrich subject and scene diversity, elevate overall data fidelity, and ensure subjects are disentangled from extraneous elements.

- **An effective image condition injection method**, dubbed R-Rope, which introduces rotary position encoding for subject tokens to maximize the model's ability to integrate information from multiple reference images. This mechanism improves multi-image and multi-subject S2V consistency, while maintaining computational efficiency.

- **A state-of-the-art (SOTA) open-source S2V model.** Extensive experiments show that our approach achieves excellent S2V performance in terms of subject fidelity, background disentanglement, and general generation quality, substantiating its effectiveness for building general-purpose, subject-consistent video generation models.

Furthermore, we will **open-source** both our **data pipeline** and **pretrained S2V model** to support the community and provide a strong foundation for future research on subject-to-video generation.

## 2 RELATED WORK

Reference-guided generation has been widely studied in both image and video domains. In the image domain, methods such as DreamBooth (Ruiz et al., 2023) explored personalized generation by fine-tuning diffusion models on a small set of reference images, enabling the preservation of subject-specific characteristics. Extensions such as IP-Adapter (Ye et al., 2023) further enhanced reference conditioning by leveraging multiple input images and contextual information, although these works remain limited to static image synthesis. In parallel, video generation has undergone a rapid evolution: GAN-based approaches (Goodfellow et al., 2014) initially struggled with stability and temporal coherence, while diffusion models based on U-Net architectures introduced notable improvements in quality. More recently, Diffusion Transformers have brought substantial progress in controllability, text alignment, and long-range temporal consistency, giving rise to a variety of downstream tasks including video editing, video inpainting, and text-to-video generation.

Building upon these advances, the subject-to-video (S2V) task has emerged as a natural extension of reference-guided generation. Proprietary systems such as Vidu (Bao et al., 2024) and Kling (Kuaishou) demonstrated the feasibility of generating videos from reference images, attracting significant attention but limiting community access due to their closed-source nature. The subsequent release of open-source frameworks such as VACE (Jiang et al., 2025) , Phantom (Liu et al., 2025) and SkyReels-A2 (Fei et al., 2025) accelerated research in this field, enabling applications ranging from digital human generation to virtual try-on and face-swapping. Despite these developments, existing S2V models (Jiang et al., 2024; Zhou et al., 2024; Wang et al., 2024) still face persistent challenges. In particular, many approaches rely on directly concatenating reference embeddings with video latents, which often leads to insufficient background disentanglement and degraded subject fidelity. When the reference subject appears in complex backgrounds, models frequently carry over background artifacts into the generated video. Moreover, in multi-subject or multi-image settings, the lack of dedicated mechanisms for reference alignment commonly results in token disorder and weakened temporal consistency.

Our work builds on these foundations while addressing the limitations of prior S2V approaches. By introducing a more comprehensive data pipeline and improved training strategies, we aim to enhance both background disentanglement and subject fidelity, moving toward a more general and robust framework for subject-aware video generation.

# 3    DATASET CONSTRUCTION PIPELINE

## 3.1    MOTIVATION

Subject-to-Video (S2V) generation addresses the challenge of synthesizing videos of a specific subject, conditioned on reference images and textual prompts. Achieving high-quality S2V in broad open-world scenarios relies on the availability of training data that is diverse in content and consists strictly of decoupled image-video pairs. In such pairs, the visual attributes of the subject must be independent of the surrounding context.

Previous work primarily utilizes datasets designed for subject-driven image generation or limited video tasks, which do not adequately meet these requirements. This leads to three significant limitations: a lack of subject and scene diversity hinders generalizability; inconsistent annotation quality decreases controllability; and image-video pairs are entangled with background information. Consequently, current S2V models often rely on separate segmentation or subject extraction steps during inference, preventing true end-to-end subject conditioning and restricting compositional flexibility.

To address these constraints and facilitate the development of genuinely end-to-end S2V models applicable in unconstrained scenarios, we propose a new dataset construction pipeline. Our approach incorporates robust grounding and segmentation, along with advanced filtering techniques and a cross-paired composition strategy designed to enforce subject-background disentanglement at scale. This process produces diverse, high-quality data pairs essential for training models capable of directly synthesizing videos from unsegmented reference images and flexible prompts, thus advancing S2V research towards practical deployment in open-domain applications.

## 3.2    PIPELINE DESIGN

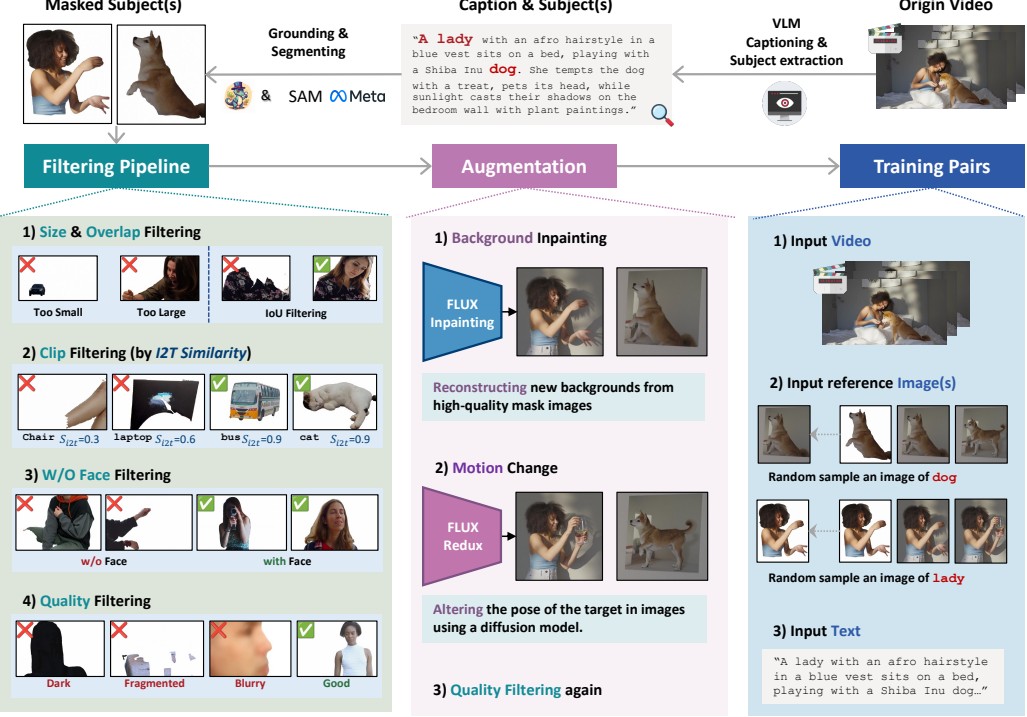

Figure 3: Scalable multi-stage data pipeline for subject-to-video (S2V) generation, where data augmentation enables the creation of cross-paired samples for robust training.

To achieve these objectives, we propose a scalable, multi-stage dataset construction pipeline 3 tailored to the needs of S2V. The process is summarized as follows:

**(1) Video Preprocessing and Captioning.** We start by slicing large-scale raw video collections into shorter clips, each containing coherent actions or events. An automatic captioning model generates textual descriptions for each clip, ensuring alignment between the visual and textual modalities.

**(2) Subject Category Definition and Candidate Identification.** To further enhance diversity, we construct a broad taxonomy of subject categories covering various domains (e.g., humans, objects, backgrounds). This taxonomy comprises over 100 distinct subject categories and includes over 800 candidate synonyms and instances, enabling captions to be matched against a rich vocabulary to identify candidate subjects ($class_v$) for further processing. This method enables scalable subject discovery without manual annotation, thus enriching the dataset with diverse subjects.

**(3) Grounding and Segmentation.** For accurate localization of subject regions, we employ Grounding DINO (Liu et al., 2024) for robust localization and SAM (Kirillov et al., 2023) for fine-grained segmentation. This combination ensures both semantic correctness and boundary precision, which are essential for effective subject-centric video generation.

**(4) Filtering and Validation.** To guarantee data quality, we implement several filtering strategies: (i) **Size Filtering:** Removes excessively small or large instances; (ii) **CLIP-Based Classification:** Verifies category alignment against textual descriptions; (iii) **IoU-Based Filtering:** Excludes instances with significant overlapping regions, ensuring distinct subject representation; and (iv) **Quality Checks:** Brightness and blur assessments filter out low-quality samples. For human categories, we use InsightFace to retain only instances with valid frontal faces, enhancing identity preservation.

**(5) Augmentation via Background Disentanglement.** One of the key challenges in S2V is the entanglement of subject and background. To mitigate this issue, we apply inpainting techniques (Labs et al., 2025) to segmented regions in the reference images, effectively erasing background information. During training, the model is encouraged to reconstruct subject appearances from the reference images while relying on textual prompts for background synthesis. This strategy prevents overfitting to incidental background cues and enhances subject transferability across various scenes.

**(6) Augmentation via Pose and Motion Enrichment.** Finally, to improve diversity and avoid overfitting to frame-level similarity, we utilize Flux Redux (Labs et al., 2025) to enrich reference images with novel poses and motions not present in the original video frames. This enhancement encourages the model to learn a more generalizable representation of subject identity that is robust to motion variations.

The resulting pipeline not only yields high-quality, background-independent subject annotations, but also provides a versatile framework applicable to a wide range of downstream applications. More importantly, it establishes a unified perspective on S2V, laying the groundwork for future research focused on subject-specific personalization and multi-task unification.

## 4 FRAMEWORK

We explore an innovative framework for video generation based on diffusion models. Our primary focus is on integrating multiple reference images to improve the video generation process. Given a set of reference images $I_1, I_2, \ldots, I_n$, a textual input $T$, and the target video $V$, our objective can be formulated as follows.

$$V = \mathcal{G}(I_1, I_2, \ldots, I_n, T; z) \tag{1}$$

where $z$ represents the stochastic noise variable intrinsic to the video generation process. The goal is for $V$ to be visually coherent, effectively encapsulating information from the reference images and adhering to textual guidance.

### 4.1 PRELIMINARIES

Text-to-video (T2V) synthesis extends diffusion-based modeling into the spatio-temporal domain, aiming to transform Gaussian noise variables $\epsilon \sim \mathcal{N}(0, I)$ into coherent videos $\mathbf{x}_0$ that correspond

to a textual description. Modern approaches employ latent diffusion schemes, where a spatio-temporal autoencoder $E(\cdot)$ compresses videos into a compact latent tensor $Z \in \mathbb{R}^{T \times C \times H \times W}$, with $T$ denoting the temporal dimension, $H \times W$ the spatial resolution, and $C$ the channel dimension. Within this latent space, transformer-based denoising networks $v_\theta$ iteratively remove noise, leveraging spatio-temporal self-attention and relative positional encodings (e.g., 3D Rotary Positional Embeddings) to capture long-range dependencies. Text conditioning is incorporated by encoding prompts through a pretrained encoder $\tau_\theta(y)$, with fusion between latent video and text features via cross-attention layers. For training, we adopt Flow Matching (Esser et al., 2024), where noise is injected as $x_t = (1-t)x_0 + t\epsilon$, and the model is optimized with the loss.

$$\mathcal{L} = \mathbb{E}_{\mathbf{x}_0, t, y, v} \left[ \|v - v_\theta(E(\mathbf{x}_0), t, \tau_\theta(y))\|_2^2 \right], \tag{2}$$

where $t$ is the diffusion timestep, and $v = \frac{dx}{dt} = \epsilon - x_0$.

## 4.2 METHOD

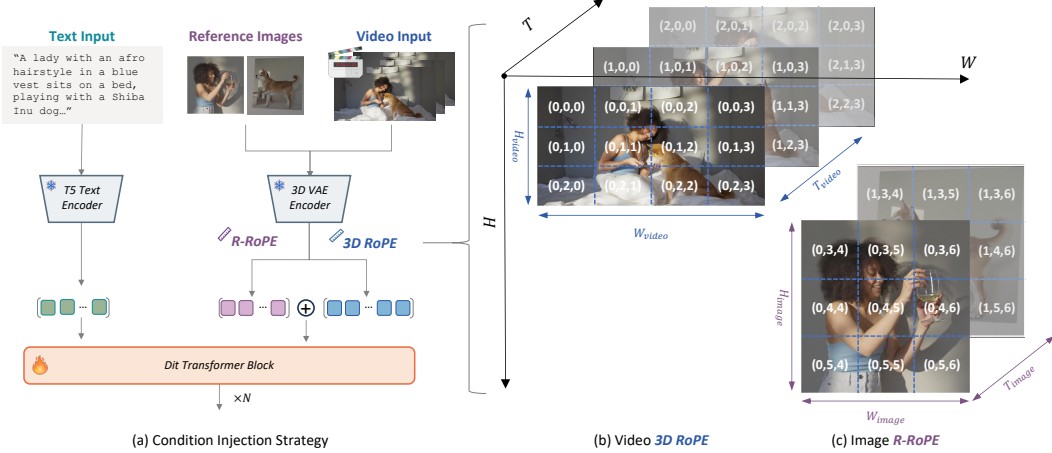

(a) Condition Injection Strategy     (b) Video **3D RoPE**     (c) Image **R-RoPE**

Figure 4: Illustration of our subject-to-video framework. (a) Multiple reference images are injected for guided video generation. (b) Video tokens use 3D RoPE positional encoding, while (c) reference images utilize **R-RoPE** for distinct spatial-temporal positioning.

In this work, we adopt a straightforward condition injection strategy to combine image conditions with video sequences for S2V tasks. Rather than employing complex adapter-based modules, we utilize a simple concatenation scheme to merge the encoded image conditions and video noise representations along the sequence dimension. Specifically, the input sequence is formulated as:

$$X = [I_1, I_2, \cdots, I_n, z] \tag{3}$$

This approach preserves the inherent structure of the original base model and enables efficient, stable learning by minimizing architectural modifications. However, direct concatenation with adjacent position ids introduces a new challenge: the model may misinterpret image conditions as consecutive frames within the video sequence, potentially disrupting temporal continuity and degrading the quality of the generated video. To address this, **it is essential for the model to differentiate image tokens from video tokens and fully understand their respective roles.**

To facilitate this distinction, we introduce the **Reference Rotary Positional Encoding (R-RoPE)** mechanism. As illustrated in Figure 4, conventional 3D RoPE encodes video tokens using positional vectors in the form $(t, h, w)$, where $t$ is the temporal frame index and $h, w$ denote spatial dimensions, each starting from zero. For image conditions, we modify the positional vectors so that their spatial dimensions are shifted to start from the maximum observed dimensions of the video sequence $(H_{max}, W_{max})$, ensuring that image tokens occupy distinct positions and are easily separable from video tokens within the model's spatial-temporal embedding space. Furthermore, the temporal positions of image conditions are individually assigned, beginning from $t = 0$ for each image. Formally, the positional vectors for each image $I_i$ is defined as:

$$\text{Pos}_i = [i - 1, H_{max} : shiftH, W_{max} : shiftW] \tag{4}$$

| | Vidu Q1 | Kling | VACE | Phantom | SkyReels | Kaleido |
|---|---|---|---|---|---|---|
| **Model Type** | *Closed-source* | | *Open-source* | | | |
| *General Video Quality Metrics* | | | | | | |
| Subject Consistency | **0.956** | 0.925 | 0.927 | 0.946 | 0.847 | **0.956** |
| Background Consistency | **0.956** | 0.940 | 0.934 | 0.952 | 0.892 | 0.953 |
| Motion Smoothness | **0.993** | 0.992 | 0.988 | 0.989 | 0.985 | 0.991 |
| Aesthetic Quality | 0.654 | 0.636 | 0.617 | 0.614 | 0.524 | **0.662** |
| Imaging Quality | 0.695 | 0.695 | 0.709 | **0.719** | 0.621 | 0.718 |
| *Text-Alignment Metric* | | | | | | |
| ViClip Score | 0.230 | 0.230 | 0.218 | **0.231** | 0.198 | 0.226 |
| *S2V-Specific Metrics* | | | | | | |
| S2V Decoupling | 0.317 | 0.316 | 0.284 | 0.308 | 0.247 | **0.319** |
| S2V Consistency | 0.704 | 0.696 | 0.710 | 0.697 | 0.682 | **0.723** |

Table 1: Quantitative results on S2V generation. Our Kaleido achieves performance across general metrics. On task-specific metrics, **S2V Consistency** and **S2V Decoupling**, Kaleido attains the best scores, demonstrating superior subject preservation and irrelevant information disentanglement.

where $i$ indexes the image conditions, and $\mathrm{shiftH}, \mathrm{shiftW}$ indicate the sum of $H_{\max}$ and the height of the reference image, and the sum of $W_{\max}$ and the width of the reference image, respectively. This explicit separation in the positional encoding prevents mixing of spatial relationships between the video sequence and the injected image conditions. By leveraging this concatenation-based condition injection and positional encoding, our model distinguishes between video and image information and generates consistent, high-quality video output within the Diffusion Transformer architecture.

## 5 EXPERIMENTS

### 5.1 IMPLEMENTATION DETAILS

Our model is fine-tuned from Wan2.1-T2V-14B through a two-stage training paradigm. The pre-training stage uses **2M** pairs for 10K steps with a learning rate of 1e-5 and batch size of 256, followed by supervised fine-tuning (SFT) on **0.5M** pairs for 5K steps with the learning rate reduced to 5e-6. Training is performed with the AdamW optimizer (Kingma & Ba, 2015), leveraging Fully Sharded Data Parallel (FSDP) and Sequence Parallelism to maximize efficiency.

### 5.2 EVALUATION METRICS

To comprehensively evaluate S2V generation, we consider metrics in three aspects. For overall video quality, we use five standard measures from VBench (Huang et al., 2024): subject consistency, background consistency, motion smoothness, aesthetic quality, and imaging quality. Semantic alignment with prompts is assessed using the ViCLIP (Wang et al., 2023) score.

We further introduce two dedicated metrics for reference-image correspondence: **S2V Consistency**: measures how well the subject identity from the reference images is preserved in the generated video. Subjects are detected and segmented using Grounding Dino and Segment Anything; CLIP features are extracted, and for each frame the maximum similarity with reference images is computed and averaged. **S2V Decoupling**: evaluates the model's ability to disentangle background information. The subject regions are masked out in both reference images and video frames; CLIP features are extracted from the backgrounds, and the score is defined as $1 - \mathrm{similarity}$ (higher is better).

To ensure robust evaluation, we construct a diverse test set covering humans, animals, cartoons, and objects, including 400 high-quality reference images and over 170 multisubject cases. These metrics together provide a comprehensive and balanced assessment of our model's performance.

### 5.3 MAIN RESULTS

**Quantitative Results** Table 1 summarizes the quantitative comparison of our method against both closed-source and open-source models. Across the five conventional metrics adopted from VBench,

our model achieves competitive performance, particularly excelling in *Subject Consistency* and *Aesthetic Quality*. In terms of motion smoothness, background consistency, and imaging quality, our method closely matching top closed-source models.

For task-specific metrics, our method demonstrates clear advantages. We obtain the highest score on *S2V Consistency* (0.723) and *S2V Decoupling* (0.319), indicating that our model more faithfully preserves subject identity from reference images while better disentangling background information. These results highlight the effectiveness of our data construction strategy and architectural design for handling subject-to-video generation.

In addition to automatic metrics, we conduct a user study evaluating four key aspects: Video Quality, Prompt Alignment, S2V Consistency and S2V decoupling. As illustrated in Figure 2, human raters consistently prefer our *Kaleido* model over both open-source and closed-source models. Notably, Kaleido achieves the highest ratings in *Video Quality*, *S2V Consistency*, and *S2V Decoupling*, further confirming the superiority of our approach from a human-centric perspective.

**Qualitative Results** As shown in Figure 5, we present qualitative comparisons across several repre-

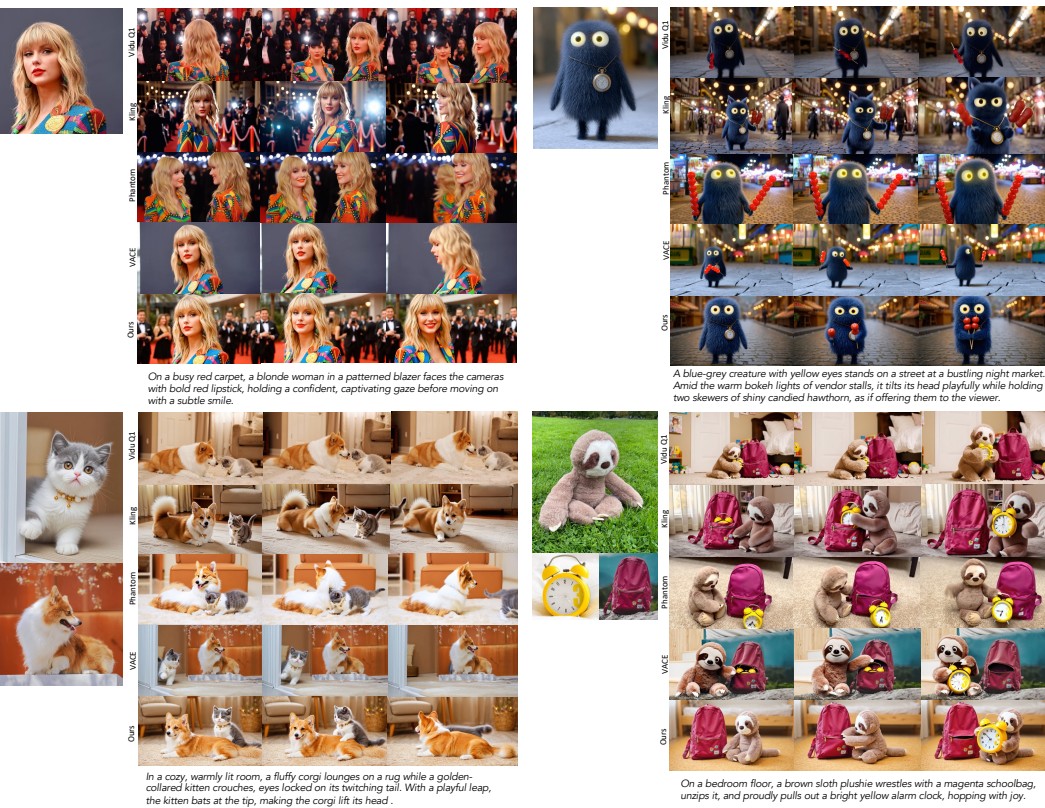

Figure 5: Qualitative comparisons , *Kaleido* clearly demonstrates superior capabilities on **S2V Decoupling**, **S2V Consistency** and **Video Quality**.

sentative scenarios. The results reveal that VACE struggles to disentangle irrelevant information, as background elements from the reference images consistently appear in the generated videos. Vidu, on the other hand, occasionally introduces redundant repetitions of reference images, causing certain subjects to appear multiple times within the generated video. Phantom exhibits similar issues and further suffers from slightly lower overall video quality. In contrast, both Kling and our model outperform other approaches in terms of S2V consistency and disentanglement of irrelevant information. However, Kling occasionally produces errors in reference fidelity—for example, in the animal case, where a small dog is incorrectly rendered with a bell around its neck. Overall, our method achieves a more balanced performance across multiple dimensions, demonstrating substantially stronger disentanglement capability while attaining S2V Consistency comparable to that of closed-source models.

|  | Concat | ShiftW | ShiftH | ShiftW&H |
|---|---|---|---|---|
| S2V Consistency | 0.661 | 0.679 | 0.687 | **0.708** |
| S2V Decoupling | 0.296 | 0.297 | 0.304 | **0.310** |

Table 2: Ablation: Comparison of R-RoPE Positional Encoding Variants.

|  | w/ Cross-Paired | w/o Cross-Paired |
|---|---|---|
| S2V Consistency | **0.708** | 0.670 |
| S2V Decoupling | **0.310** | 0.297 |

Table 3: Ablation: Impact of Cross-Paired Data Inclusion.

### 5.4 ABLATION STUDY

We conduct comprehensive ablation studies to evaluate the effects of key components in our methodology. Specifically, we analyze the impact of cross-paired data construction on subject-relevant disentanglement and the influence of our proposed R-RoPE positional encoding strategy.

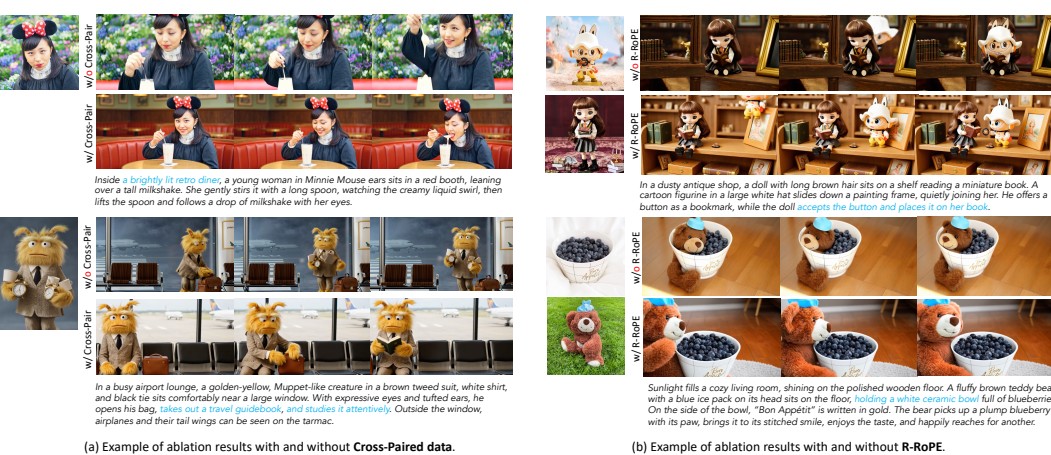

(a) Example of ablation results with and without **Cross-Paired data**.

(b) Example of ablation results with and without **R-RoPE**.

Figure 6: Ablation visualizations. (a) Effect of Cross-Paired data. (b) Effect of R-RoPE.

**Effect of Cross-Paired Data Construction.** To assess the contribution of Cross-Paired data construction to disentangling subject-specific and irrelevant information, we compare models trained with and without this data. As shown in Table 3, excluding Cross-Paired data leads to a notable decrease in both S2V Consistency and S2V Decoupling metrics, demonstrating its effectiveness. Figure 6a further illustrates that Cross-Paired data training enables the model to better separate the subject from unrelated elements, such as backgrounds and handheld objects. This promotes generation of diverse backgrounds while maintaining focus on the subject, thus facilitating robust decoupling between subject and irrelevant information representations.

**Effect of R-RoPE Positional Encoding.** We further ablate the proposed R-RoPE positional encoding by considering four settings: (1) baseline (without R-RoPE), (2) shifting only width (ShiftW), (3) shifting only height (ShiftH), and (4) shifting both width and height (ShiftWH). As reported in Table 2, simultaneous spatial shifts result in the highest subject consistency and irrelevant information decoupling. Furthermore, Figure 6 demonstrates that R-RoPE mitigates reference confusion and subject overlap in multi-subject scenarios. These findings confirm that our R-RoPE design is essential for enhancing multi-reference integration and preventing reference-token misalignment during generation.

## 6 CONCLUSION

In this work, We introduced ***Kaleido***, a subject-to-video generation framework to addresses the challenges of multi-subject consistency and background disentanglement under multi-image conditioning. By constructing a diverse and high-quality training dataset with cross-paired samples, and by proposing the R-RoPE positional encoding strategy, Kaleido achieves superior subject preservation and irrelevant information separation, outperforming open-source models and approaching closed-source models.

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

# A APPENDIX

## A.1 DATA PIPELINE AND DATASET DETAILS

Our data pipeline comprises two essential stages:

**(1) High-quality Subject Segmentation:** We first perform robust subject segmentation to obtain precise masks of target subjects from raw video frames. Only those instances passing stringent quality criteria (size, clarity, semantic alignment) are retained for downstream processing.

**(2) Cross-Paired Sample Construction:** Rather than using segmented subjects with their original backgrounds, we construct **cross-paired** data samples. In this process, segmented subjects are composited with backgrounds or video clips from different, unrelated sources. This ensures that the model learns to disentangle subject identity from background and pose, thereby preventing it from overfitting to trivial copy-paste solutions or memorizing original contexts.

To further ensure dataset quality, we conducted a manual evaluation of the constructed dataset (segmentation subset). The results were favorable, owing to the use of high-confidence thresholds during the automated filtering stage — including strict localization thresholds in Grounding DINO, high CLIP similarity thresholds for category filtering, and high-confidence face detection for human subjects. The evaluation shows that the CLIP-based category filter achieves a success rate of over $92\%$, and face detection exceeds $95\%$ accuracy, confirming the reliability of our pipeline.

For model robustness, we adopt a balanced training data ratio:

$$\text{crop} : \text{segment} : \text{inpainting} : \text{redux} = 1 : 5 : 3 : 1$$

where **crop** refers to cropped subjects, **segment** indicates high-quality segmentation, **inpainting** means removing backgrounds via generative inpainting, and **redux** refers to pose/motion augmentation. Importantly, we find that inpainting and redux samples serve primarily as guidance, so their proportions are kept moderate.

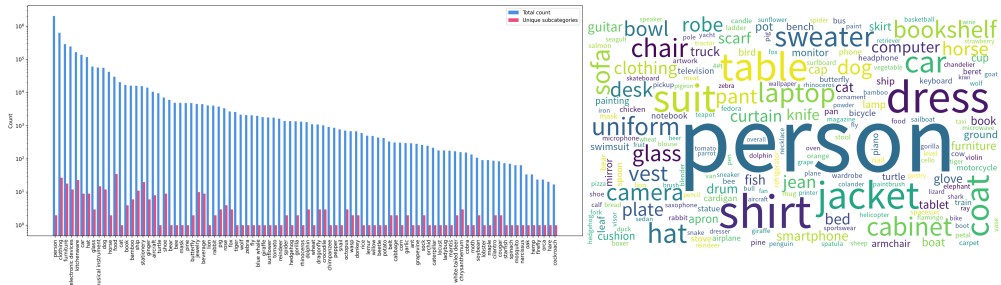

Figure 7: Statistics of our training dataset. .

**Role of Cross-Pair Data.** As illustrated in Fig. 6, training with only crop or segment inputs easily leads the model to memorize reference-image backgrounds and poses, resulting in copy-paste–like artifacts. Cross-paired samples mitigate this issue by explicitly breaking the correlation between subject identity and its original context, thereby enforcing stronger subject–background disentanglement and improving generalization.

To further quantify this effect, we conduct an ablation study on the proportion of cross-paired samples in training. As shown in Table A.1, introducing a moderate amount of cross-pairing (20–40%) significantly improves both *consistency* and *decoupling*. However, excessively large ratios (e.g., 80%) begin to degrade performance, suggesting that cross-pair data act as a regularizer: insufficient cross-pairing fails to eliminate context coupling, whereas overly aggressive cross-pairing suppresses high-fidelity subject signals required for S2V generation.

In addition, while **inpainting** and **redux** samples provide valuable supervisory signals—such as improving motion robustness and enhancing background flexibility—their contributions saturate when used excessively. Therefore, maintaining these samples at moderate proportions is crucial for preserving the balance between subject fidelity and generalization.

| Cross-Pair Ratio | Consistency | Decoupling |
|:---:|:---:|:---:|
| 0% | 0.670 | 0.297 |
| 20% | 0.692 | 0.306 |
| 40% | **0.708** | 0.310 |
| 60% | 0.673 | **0.312** |
| 80% | 0.664 | 0.307 |

Table 4: Impact of cross-pair ratios on subject consistency and subject–background decoupling.

### A.2 DESIGN AND ANALYSIS OF R-RoPE

**Motivation.** Our goal is to investigate how a subject-to-video (S2V) model can more efficiently learn *extra visual conditions*—multiple reference images—during video generation. Existing approaches generally fall into two categories. (1) **Adapter-based methods** (e.g., VACE), which insert additional transformer layers for conditioning. Achieving competitive performance requires stacking many layers, introducing substantial increases in FLOPs and parameter count (e.g., adding $0.5\times$ parameters to a 1.3B model). (2) **Sequence-concatenation methods** (e.g., Phantom), which keep base model parameters frozen and inject reference information by directly appending image tokens along the sequence dimension, thus retaining computational efficiency.

Motivated by these observations, we adopt *Sequence-Concatenation* for injecting subject conditions. We explored both *Channel-Concatenation* and *Sequence-Concatenation*. Although channel concatenation works well in I2V settings due to spatial alignment between image and video frames, it is less suitable for S2V, where alignment relies on semantic correspondence rather than spatial correspondence. Consistent with this hypothesis, our experiments(Table 6) show that channel concatenation and produces noticeably inferior S2V subject fidelity.

However, direct sequence concatenation introduces a clear issue: when reference images are appended as the last few frames, the model exhibits **subject overlap artifacts** (see Fig. 6), i.e.. This indicates that treating images as ordinary video frames disrupts the temporal generation prior. These observations motivate designing a positional encoding mechanism that explicitly distinguishes between *video tokens* and *reference-image tokens*. Rotary Position Embedding (RoPE) naturally provides the necessary degrees of freedom.

**Design of R-RoPE.** We extend RoPE by applying deterministic shifts along the spatial (H/W) and temporal (T) index only to reference-image tokens, while keeping video tokens untouched. The design follows three principles: (1) video and image tokens must be easily distinguishable; (2) different reference images should be separable; (3) no artificial temporal continuity should be implied among images.

**Evaluated Variants.** To systematically examine which indices carries meaningful conditioning signals, we evaluate the following R-RoPE variants:

| Method | Subject | Background |
|:---|:---:|:---:|
| ShiftW&H | **0.708** | **0.310** |
| ShiftW&H w/o overlap | 0.690 | 0.296 |
| Staggered Negative Time Shift (T = -3, -2, -1) | 0.683 | 0.276 |
| Fixed-Time Encoding (T = -1) | 0.644 | 0.298 |
| Future-Shifted Time Encoding (T = +1, +2, +3) | 0.667 | 0.300 |
| Channel-Concat | 0.632 | 0.285 |

Table 5: Extended evaluation of R-RoPE variants for S2V conditioning. Metrics correspond to subject consistency (left) and background consistency (right).

- **Spatial-only variants**: **ShiftW&H**: all reference images share identical spatial shifts. **ShiftW&H w/o overlap**: each reference image receives an independent spatial shift, removing overlap in the (H,W) RoPE indices.

- **Temporal-shift variants**: We define three families of temporal manipulations, each corresponding to how the $t$-indices of reference images are assigned: Here, $T$ denotes the pseudo-time index along the temporal dimension, with negative values representing positions before the video sequence, and positive values after it.

    – **Fixed-Time Encoding (T = -1)**: for each reference subject, all associated images are assigned a fixed negative time index $T = -1$.

    – **Staggered Negative Time Encoding (T = -3, -2, -1)**: for each reference subject, the images are assigned progressively earlier negative time indices, e.g., $T = -3, -2, -1$.

    – **Future-Shifted Time Encoding (T = +1, +2, +3)**: for each reference subject, the images are assigned time indices following the final frame of the video sequence, e.g., $T = N+1, N+2$, where $N$ is the number of video frames.

**Results.**    Table A.2 summarizes the quantitative performance for all newly evaluated variants. The **spatial-only R-RoPE with shared H/W shifts** yields the best subject fidelity and background consistency. In contrast, **temporal shifts consistently underperform**: either fixing the time index, assigning staggered negative indices, or shifting references into future frames all introduce modality bias that degrades the model's ability to learn subject semantics.

**Summary.**    Overall, the extended analysis reveals: (1) S2V requires semantic-level conditioning, making sequence concatenation more suitable than channel concatenation; (2) positional embeddings are critical to prevent reference-image duplication artifacts; (3) **spatial-only R-RoPE with shared H/W shifts is the most stable and effective design**, while temporal shifts introduce modality inconsistency and consistently degrade performance.

