# OpenReview forum: "Kaleido: Open-Sourced Multi-Subject Reference Video Generation Model"
_ICLR.cc/2026/Conference — Submitted to ICLR 2026_

### Official Review · Reviewer_45Cb · 2025-10-29

**Soundness:** 3
**Presentation:** 3
**Contribution:** 2
**Rating:** 4
**Confidence:** 5

**Summary:**

This paper proposes Kaleido, a multi-subject reference video generation model， aiming to address the shortcomings of existing S2V models in maintaining multi-subject consistency and background disentanglement. Its core innovations include: constructing a dedicated training data pipeline with low-quality sample filtering and cross-paired data synthesis, and proposing Reference Rotary Positional Encoding (R-RoPE) to achieve stable and accurate multi-image integration. Experiments show that Kaleido significantly outperforms existing open-source models in key metrics such as subject consistency  and background disentanglement and is comparable to closed-source models

**Strengths:**

- R-RoPE is an effective strategy:  Isolating reference images only in the temporal  dimension is indeed unreasonable. This approach is prone to confusing the model between reference image information and the original video generation sequence during video generation, as the model may misinterpret reference images as consecutive frames in the video. Therefore, it is necessary to additionally introduce distinctions in the width (W) and height (H) spatial dimensions.

**Weaknesses:**

1. Lack of innovation in the data pipeline: In related methods such as Conceptmaste and  Phantom-data [1,2], in-depth explorations have been conducted on cross-paired strategies for synthetic data and real data.
2. Lack of experimental validation for data pipeline-related methods: The paper only provides an overall comparison of experimental results, it lacks targeted validation for individual key components of the pipeline.

[1] ConceptMaster: Multi-Concept Video Customization on Diffusion Transformer Models Without Test-Time Tuning
[2] Phantom-Data : Towards a General Subject-Consistent Video Generation Dataset

**Questions:**

None

---

> ### Author Response · Authors · 2025-11-24
> **Response to Reviewer 45Cb**
>
> We thank the reviewer for your detailed comments and constructive feedback on both the strengths and areas of improvement.
>
> ---
>
> **Weaknesses 1:** Lack of innovation in the data pipeline: In related methods such as Conceptmaster and Phantom-data, in-depth explorations have been conducted on cross-paired strategies for synthetic data and real data.
>
> Thank you for the comparison with Conceptmaster and Phantom-data. We would like to clarify a few differences in our contribution:
>
> - In **Conceptmaster**, the emphasis is on obtaining high-quality segmentation and precise concepts, but it does not specifically address cross-pair construction for background disentanglement. Their approach does not enable full background–subject decoupling as is required for the S2V task.
>
> - **Phantom-data** is a contemporaneous work with ours. Their cross-pair samples are built through a *cross-frame similarity* search combined with *visual-language model (VLM) filtering*, which ensures diversity but involves a relatively complex pipeline. Moreover, this process requires long-video sources (e.g., movies or TV series) to observe the same subject in different scenes, imposing additional constraints on data availability and quality.
>
> In contrast, our cross-pair strategy is both **simple** and **effective**: it does not depend on specific source types or lengthy videos, and directly uses segmentation masks with synthetic inpainting plus controlled Redux augmentation. This design maximizes *subject identity consistency* while maintaining scalability.By comparison, cross-pair construction based on similarity computation and VLM filtering inevitably introduces some identity mismatches.
> Our experiments (Table 4) also demonstrate that it is unnecessary to make all training samples cross-pair — a balanced proportion of around **40%** already yields optimal S2V performance.
> Quantitatively (Table 1), our method achieves superior background disentanglement and subject consistency, and overall higher S2V metrics than Phantom.
>
> ---
>
> **Weaknesses 2:** Lack of experimental validation for data pipeline-related methods: The paper only provides an overall comparison of experimental results, it lacks targeted validation for individual key components of the pipeline.
>
> Our data pipeline consists of two primary components:
> 1. **Filtering pipeline** — the construction of high-quality segmentation data.
> 2. **Augmentation** — the construction of cross-pair samples.
>
> We performed targeted validation for the cross-pair component through ablation experiments (see Figure 6 and Table 3), which show that including cross-pair data enables the model to better disentangle background information and subject motion.
> **Quantitatively**, adding cross-pair samples increases **S2V Consistency** from `0.670` to `0.708` and **S2V Decoupling** from `0.297` to `0.310`. These results confirm that cross-pair construction is an **essential part** of our pipeline, providing significant gains in both maintaining subject identity and separating it from irrelevant variations.
>
> To further address your concern regarding validation of the automated data pipeline, we conducted a detailed evaluation of the *high-quality segmentation* subset. In this process, our **Grounding DINO** localization, **CLIP-based** category filtering, and **face detection** for human subjects were all applied with *high-confidence thresholds* to ensure that the final segmented content is correct.
> Manual evaluation confirms that the CLIP-based category filter achieves a success rate of over **92%**, and face detection exceeds **95%** accuracy.  In addition, we also manually checked our cross-pair samples: since inpainting does not alter subject identity and our Redux augmentation uses small parameters to only adjust pose, the identity consistency of cross-pair data is maintained at over **99%**.
>
> These results demonstrate that our dataset construction process reliably enforces both category correctness and subject fidelity, thereby ensuring the robustness and overall quality of the proposed dataset.
>
> ---
>
> We appreciate your suggestions and have made corresponding clarifications in the revised manuscript.

---

### Official Review · Reviewer_FNr6 · 2025-10-31

**Soundness:** 3
**Presentation:** 3
**Contribution:** 3
**Rating:** 6
**Confidence:** 4

**Summary:**

This paper propose Kaleido, a fully open-sourced S2V generation model. It includes a scalable data pipeline to collect training data and a lightweighted conditioning schemes that apply R-Rope. It achieves SOTA result among open-sourced S2V models.

**Strengths:**

- The paper is well-written and easy to understand.
- The proposed data collection pipeline takes into account the cross-paired images, that can solve the background leakage problems during training.
- The proposed R-RoPE is simple but effective to disentangle denoised image from condition.
- The model achieves SOTA results, and it is fully open sourced and faciliate the community.

**Weaknesses:**

- This proposed framework concatenates tokens but not token-channels, which may make the inference slow.
- The paper does not discuss why they did not use channel-wise concatenation, which is efficient and widely adopted.
- For R-RoPE, why the t-dim of RoPE for refernces images is not shift-T?
- The paper lacks novelty and is mostly engineer work, but it should be fine.

**Questions:**

see weaknesses

**Details Of Ethics Concerns:**

human face may be used as reference images in the collected dataset, make sure it is authorized since the model will be fully open sourced

---

> ### Author Response · Authors · 2025-11-24
> **Response to Reviewer FNr6**
>
> We thank the reviewer for their positive comments on the clarity of our writing, the effectiveness of our data pipeline and R-RoPE design, and the open-source contribution. We address your technical concerns below:
>
> ---
>
> **Weaknesses 1:** This proposed framework concatenates tokens but not token-channels, which may make the inference slow.
>
> In the S2V task, the token sequence length of reference images is much shorter than that of video frames, so the additional overhead introduced by conditioning is relatively small. Our method is about **+11%** slower than the base Wan model under identical settings, yet still maintains competitive inference speed.
>
> By contrast, leading open-source models such as VACE and Phantom employ relatively complex injection strategies:
> - **VACE** uses an **adapter-based architecture**, and for the 14B model adds five extra transformer layers on top of sequence concatenation, which increases both FLOPs and inference time.
> - **Phantom** uses a **classifier-free design** that requires computing three sets of embeddings per video frame, further increasing runtime and memory use.
>
> Under the same experimental conditions (14B model, 512p, 3 reference images, Nvidia H800), our method provides competitive inference speed.
>
> **For reference:**
>
> | Method  | Wan  | Ours | VACE | Phantom |
> |---------|------|------|------|---------|
> | Average inference time (s) | 556 | 617 | 754 | 854 |
> | Slower than base Wan (%)    | --  | +11%| +36%| +54%|
>
> Thus, our efficient conditioning strategy achieves favorable performance compared to widely used alternatives.
>
> ---
>
> **Weaknesses 2:** The paper does not discuss why they did not use channel-wise concatenation, which is efficient and widely adopted.
>
> We do not use channel-wise concatenation because it is suited for spatially corresponding inputs, especially in image-to-video (I2V) generation with first-frame or key-frame guidance. In the S2V task, the priority is to learn the *identity information* of reference images rather than directly copying spatial positions. Moreover, in multi-reference S2V, stacking different reference images along the channel dimension can cause *temporal confusion*: clearly unrelated images end up merged with video tokens that have a defined temporal order, which may mislead the model.
>
> It is worth noting that in existing S2V work, channel-wise concatenation is rarely used; most leading approaches adopt sequence-based conditioning rather than merging along the channel dimension. In contrast, sequence concatenation keeps the semantic conditioning separate and avoids introducing false temporal relationships.
>
> Our experiments show that **channel-wise concatenation performs significantly worse than** sequence concatenation, with lower scores in both S2V Consistency and Background Decoupling, and is more difficult to train in multi-reference settings. Full details and quantitative metrics are provided in Appendix Table 5.
>
> ---
>
> **Weaknesses 3:** For R-RoPE, why the t-dim of RoPE for reference images is not shift-T?
>
> Thank you for raising this question. In the newly uploaded PDF, we provide a detailed discussion and a comprehensive set of experiments concerning different ways to apply temporal (T axis) positional shifts for reference images in R-RoPE. We systematically evaluated multiple shift-T strategies, including:
> - Fixed-Time Encoding ($T=-1$)
> - Staggered Negative Time Shift ($T=-3, -2, -1$)
> - Future-Shifted Time Encoding ($T=+1, +2, +3$)
>
> We assessed both quantitative metrics and qualitative outputs. The results, summarized in Table 5 of the appendix, consistently show that **all shift-T variants perform worse** than the spatial-only shift solution. We attribute this to the fact that introducing a T-dimension shift can confuse the model by implying a temporal relationship that does not exist between reference images and video frames, negatively impacting conditioning effectiveness. For this reason, and as evidenced by our thorough ablation study in the appendix, we adopted purely spatial shifts and do not use shift-T in our final design.
>
> Thank you again for your constructive feedback.

---

### Official Review · Reviewer_LNtv · 2025-11-01

**Soundness:** 3
**Presentation:** 3
**Contribution:** 2
**Rating:** 4
**Confidence:** 4

**Summary:**

This paper presents Kaleido, an open-source framework for subject-to-video (S2V) generation that focuses on maintaining multi-subject consistency and background disentanglement. The authors propose (1) a comprehensive data construction pipeline with cross-paired data, filtering, and augmentation; and (2) a novel Reference Rotary Positional Encoding (R-RoPE) for integrating multiple reference images. Experiments show that Kaleido achieves state-of-the-art results on both general video quality metrics and S2V-specific metrics, approaching the performance of closed-source systems like Kling and Vidu.

**Strengths:**

1.The proposed large-scale, cross-paired data construction process is well-designed and will be valuable for the community.
2. Comprehensive experiments: Evaluation covers humans, objects, and multi-subject settings, with both quantitative and user studies.

**Weaknesses:**

1. The architectural novelty is limited. The model mainly relies on simple concatenation for conditioning; R-RoPE, while useful, is a modest modification. Besides, its design is mostly empirical without deeper analysis.
2. The validation of proposed dataset is missing. It lacks quantitative evidence for dataset diversity and annotation accuracy, as well as the comparision with previous dataset.

**Questions:**

None

---

> ### Author Response · Authors · 2025-11-24
> **Response to Reviewer LNtv (1/N)**
>
> We sincerely thank the reviewer for recognizing the strengths of our data construction pipeline and our comprehensive experimental evaluation. We address your concerns below.
>
> **Weaknesses 1:** The architectural novelty is limited. The model mainly relies on simple concatenation for conditioning; R-RoPE, while useful, is a modest modification. Besides, its design is mostly empirical without deeper analysis.
>
> We would like to clarify that our conditioning strategy and R-RoPE design are the result of **systematic exploration** rather than an ad-hoc empirical adjustment.
> In S2V generation, existing conditioning approaches are typically either **adapter-based** (e.g., VACE) or **concatenation-based** (e.g., Phantom). We chose **sequence concatenation** for its efficiency and minimal impact on the base model, since channel-wise concatenation is less suitable when reference images and video frames do not share spatial alignment (Appendix Table&nbsp;5).
>
> On top of this, we investigated how positional encoding affects multi-reference conditioning.
> Standard **3D RoPE** decomposes positions into **temporal** ($t$), **height** ($h$), and **width** ($w$) components, enabling the base model to jointly model sequential cues, spatial structure, and long-range video coherence.
> However, in S2V the reference images are not temporally continuous with the target video: using unmodified 3D RoPE assigns them overlapping $(t, h, w)$ coordinates with video tokens, causing the model to misinterpret them as part of the generated clip, which leads to *reference confusion* and degraded subject–background disentanglement.
>
> To solve this, our **R-RoPE** retains the 3D decomposition but spatially shifts the $(h, w)$ indices of all reference-image tokens beyond the maximum video coordinates ($H_{\mathrm{max}}, W_{\mathrm{max}}$), while giving each image its own local temporal index.
> This explicit separation in the RoPE embedding space allows the model to distinguish reference tokens from video tokens while preserving the original 3D RoPE’s ability to capture spatio-temporal relations.
>
> As shown in our Appendix (*Design and Analysis of R-RoPE*), we systematically analyzed various RoPE formulations, with a particular focus on temporal-shift (RoPE-T) variants, and compared them against spatial-only designs.
> Experiments confirmed that spatial-shifted R-RoPE consistently yields the highest **S2V Consistency** and **S2V Decoupling** scores, achieving effective multi-image integration with no additional parameters or inference cost (see Table&nbsp;5).

---

> ### Author Response · Authors · 2025-11-24
> **Response to Reviewer LNtv (2/N)**
>
> **Weaknesses 2:** The validation of proposed dataset is missing. It lacks quantitative evidence for dataset diversity and annotation accuracy, as well as the comparison with previous datasets.
>
> Thank you for this suggestion. In the revised Appendix&nbsp;A, we provide detailed statistics about our dataset’s diversity and annotation accuracy.
>
> As shown in Figure&nbsp;7, our dataset covers **100+ top-level semantic categories** and over **800 fine-grained subcategories**, spanning humans, animals, plants, daily objects, and consumer goods. This rich taxonomy ensures both diversity and deep coverage for S2V training.
>
> Regarding annotation accuracy, our dataset construction involves two key processes:
>
> 1. **High-quality segmented data construction:**
>    We apply **Grounding DINO** for object localization, **CLIP-based category filtering**, and **face detection** for human subjects, all with **high-confidence** thresholds to ensure quality.
>    Based on your suggestion, we conducted manual evaluation of our segmentation subset. Results show that DINO localization, CLIP category filtering, and face detection all achieve accuracies above **90%**, confirming the reliability of our automatic filtering pipeline.
>
> 2. **Cross-pair sample construction:**
>    Since our **inpainting** model does not alter subject identity, and the **Flux-Redux** augmentation is used with small parameters to slightly adjust pose information only, both processes preserve subject identity fidelity.
>    Manual evaluation indicate that subject identity is maintained with over **99%** accuracy in cross-pair samples.
>
> ---
>
> For comparison with previous datasets, beyond **OpenS2V** [1], we also consider **Phantom** data construction. Both OpenS2V and Phantom generate cross-pair data by *cross-frame matching*, which requires the same subject to appear in different scenes within a long video (e.g., movies or TV series). This imposes a strong constraint on available source data.
>
> In contrast, our cross-pair construction method does *not* rely on long videos: it can be applied to any type of video or even still-image sources, enabling broader scalability. While our backgrounds may be less realistic than those in OpenS2V or Phantom, our **inpainting+Redux** approach maximizes preservation of the original subject's identity consistency.
>
> Moreover, cross-frame-based methods have a non-negligible probability of generating mismatched reference images. Our experiments show that for S2V tasks, **subject consistency is the dominant condition**, and effective background disentanglement can still be achieved with inpainting-generated backgrounds that are not perfect but sufficient for training.
> Furthermore, our model achieves **higher S2V Consistency and S2V Decoupling** metrics than Phantom, further validating the effectiveness of our dataset construction approach.
>
> [1] Yuan S, He X, Deng Y, et al. *Opens2v-nexus: A detailed benchmark and million-scale dataset for subject-to-video generation*. arXiv preprint arXiv:2505.20292, 2025.
>
> We thank the reviewer again for your constructive comments. We have enhanced the manuscript and appendix to provide additional clarity and experimental evidence as suggested.

---

### Official Review · Reviewer_xLfE · 2025-11-01

**Soundness:** 3
**Presentation:** 3
**Contribution:** 3
**Rating:** 6
**Confidence:** 4

**Summary:**

This paper introduces an open-source subject-to-video (S2V) generation framework that creates subject-consistent videos from multiple reference images and text prompts. Built upon the Wan 2.1 T2V-14B base model and fine-tuned for multi-reference input, it achieves near–closed-source performance in subject fidelity, background disentanglement, and video quality.

**Strengths:**

1. Introduce a pipeline to enhance subject and scene diversity, improve overall data fidelity, and ensure clear separation of subjects from irrelevant components.
2. A reference-based position encoding to emphasize the references, leading to better results.

**Weaknesses:**

1. The paper use CLIP as evaluation metrics. However, CLIP is not finegrained enough for Subject consistency. I suggest using face recognition metrics for human faces.

**Questions:**

1. How does the artifacts produced by image editing methods like Flux affects the generated video? For example, Flux redux reposes the human, which could introduce subject inconsistencies.
2. How many subjects can be inserted to the video at the same time? What is the limiting factor?

---

> ### Author Response · Authors · 2025-11-24
> **Response to Reviewer xLfE**
>
> **Weaknesses:** The paper uses CLIP as evaluation metrics. However, CLIP is not fine-grained enough for Subject consistency. I suggest using face recognition metrics for human faces.
>
> Thank you very much for your valuable suggestion. In general S2V tasks, CLIP-based feature similarity is a widely adopted metric for assessing subject consistency. However, we agree that CLIP feature similarity is not always precise at the fine-grained level for human faces.  Following your advice, we collected the corresponding human test subset and evaluated face similarity using two face recognition metrics: FaceSim-cur and FaceSim-arc. The results are summarized below.
>
> | Model            | FaceSim-cur | FaceSim-arc | Average |
> |------------------|-------------|-------------|---------|
> | **Ours**         | _0.515_     | _0.492_     | _0.504_ |
> | VACE             | 0.421       | 0.395       | 0.408   |
> | Phantom          | 0.423       | 0.400       | 0.412   |
> | Kling (closed)   | 0.507       | 0.484       | 0.495   |
> | **Vidu (closed)**| **0.549**   | **0.521**   | **0.535** |
>
> As can be seen, our model achieves the highest face similarity among all open-source approaches, and slightly outperforms the closed-source Kling.
> We appreciate your suggestion, and we will add these face recognition results and discussion to the revised paper.
>
> ---
>
> **Q1. How do artifacts produced by image editing methods like Flux affect the generated video?**
> Our Flux-based augmentation pipeline includes two distinct operations:
> 1. **Flux-Inpainting** – replaces background regions in the reference images without altering subject identity.
> 2. **Flux-Redux** – can modify a subject’s pose or motion, but in our approach is applied with relatively small parameter settings so that changes are minor and identity features are maximally preserved.
>
> As discussed regarding cross-pair data in our paper, Flux methods enable us to build training data that combines different backgrounds and poses, which is important for background and action disentanglement. However, as you mentioned, this editing can also negatively impact subject consistency if used excessively.
> Through experimental validation (see Table 4), we found that a balanced composition of *real data*, *inpainting*, and *Flux-Redux* (with pose change control) at a ratio of **6:3:1** allows the model to maintain high subject consistency while significantly improving S2V Decoupling. Increasing the proportion of Flux-Redux samples beyond this level may lead to the loss of fine-grained subject details.
>
> ---
>
> **Q2. How many subjects can be inserted into the video at the same time? What is the limiting factor?**
> There is no limit in our model’s design; theoretically, any number of reference subjects can be integrated. In our training setup, we used up to three reference images, which provided the best consistency. During testing, we observed that even with more reference images, the model can still generate coherent results. Therefore, the main limiting factor is the number of reference images included during training. The R-RoPE positional encoding also helps accommodate more than the maximum seen during training, but if the reference count is much higher, subject consistency will decrease.

---

### Author Response · Authors · 2025-11-28
**General Response**

We sincerely thank all reviewers for taking the time to read our paper and provide thoughtful feedback.

**Positive Feedback**
Several points in our work were highlighted by different reviewers: **xLfE** appreciated our clear pipeline design and the proposed reference-based positional encoding. **LNtv** valued the large-scale cross-paired data construction and the breadth of our evaluation across multiple subject types. **FNr6** pointed out the clarity of the writing, the effectiveness of our background leakage handling, the simplicity and impact of R-RoPE, and noted the importance of making our model fully _open-sourced_ for the community. **45Cb** agreed on the need for spatial separation in R-RoPE to avoid confusion between references and generated frames, and acknowledged our improvements over existing open-source S2V models.


**Clarifications to Concerns**
Some concerns appeared across multiple reviews, mainly regarding the novelty of the architecture, dataset validation, and conditioning design.

- **R-RoPE design** (**LNtv**, **FNr6**):
  R-RoPE was developed after **systematic exploration** of positional encoding strategies for multi-reference S2V.
  Spatially shifting reference tokens beyond the video coordinate range consistently gave the best results for maintaining subject consistency and background separation, while *all tested* temporal shift (shift-T) variants reduced performance.
  Our design preserves the base model’s spatio-temporal reasoning while eliminating reference confusion, as detailed in the revised appendix. The simplicity here is intentional, arising from efficiency and effectiveness rather than lack of exploration.

- **Dataset validation and comparison** (**LNtv**, **45Cb**):
  We constructed a large and diverse training corpus via our **data processing pipeline**, spanning **100+ top-level** and **800+ fine-grained** categories. Annotation accuracy exceeds **90%** and identity preservation in cross-pair samples exceeds **99%**.  Ablation studies confirm the role of cross-pair data in improving **both identity consistency and background disentanglement** (S2V Consistency: `0.670` to `0.708`, Decoupling: `0.297` to `0.310`).

  **Regarding Phantom-data** we emphasize here that **Phantom-data and our data processing pipeline are contemporaneous works developed independently**. Phantom-data’s pipeline relies on long video sources (movies/TV series) to find multiple scenes of the same subject. This constraint limits scalability and still risks identity mismatches. Our pipeline is **source-type agnostic**, works directly from images or short clips, and uses inpainting plus controlled Redux augmentation to ensure fidelity.

  This independent design is inherently simpler, more scalable, and empirically achieves **higher S2V Consistency/Decoupling metrics** than Phantom in our evaluations. Our contributions are **not based on** Phantom or any other prior dataset pipeline.

- **Conditioning choice and efficiency** (**FNr6**):
  We use **sequence concatenation** instead of channel-wise concatenation to prevent false temporal correlations in multi-reference inputs without spatial alignment.
  Experiments show channel-wise concatenation reduces both identity and background separation scores in multi-reference S2V.
  Our approach yields better metrics, with inference only **~11% slower** than base Wan, but faster than **VACE (+36%)** and **Phantom (+54%)**.

- **Additional evaluations** (**xLfE**):
  We added **FaceSim-cur** and **FaceSim-arc** metrics for human subjects, showing our model ahead of all other open-source methods and slightly above closed-source Kling.
  We also report that a **6:3:1 ratio** of Flux-inpainting / Flux-Redux / real data is effective for retaining identity while improving background and motion diversity.


We respectfully note that the primary novelty concern raised was about the R-RoPE architecture. We have provided detailed experimental evidence and analysis showing R-RoPE to be a purposeful and impactful design, delivering measurable improvements over baseline positional encoding strategies in multi-reference S2V tasks. Our data processing pipeline demonstrates clear scalability and performance advantages without relying on long-video sources.

We believe **Kaleido** advances open-source S2V generation by:
1. Achieving performance **comparable to closed-source systems** (Kling, Vidu).
2. Providing a **validated, scalable, and independently developed data processing pipeline** for multi-subject consistency and background disentanglement.
3. Offering an **efficient conditioning strategy** with proven benefits in multi-reference scenarios.
4. **Fully open-sourcing** the Kaleido model, including both training and inference, to benefit the research community.

We thank the reviewers again for their constructive feedback and hope this consolidated response clarifies our position and the distinctive contributions of our work.

---

### Meta-Review · Area_Chair_ND22 · 2025-12-17

**Summary:**

This paper proposed a Multi-Subject Reference Video Generation Model by using the proposed R-RoPE design.

Two reviewer give 6 and 2 gives 4.
In general,
1. Concerned about the novelty since R-RoPE design is more a concatenation method with sequence concatenation.
2. Lack of experimental validation for data pipeline-related methods.
3. The validation of proposed dataset is missing



Though authors did address most of the concerns raised by reviewer. But in general, I do agree that R-RoPE design is not novel enough to make it for ICLR. In addition, Lack of experimental validation for data pipeline-related methods from original submission is a sign for rejection.

**Reviewer Concerns:**

For concerns1 , authors do give a more comprehensive  response to express that's not a ad-hoc desing.
For concern2, authors add more experimental results in the rebuttal.
For concern3, authors also add n the revised Appendix A

**Reviewer Scores:**

two with 6
two with 4

---

### Decision · Program_Chairs · 2026-01-26

Reject